# XYZ Data Efficiency: Improving Deep Learning Model Quality and Training Efficiency via Efficient Data Sampling and Routing

## Abstract

Recent advances on deep learning models come at the price of formidable training cost. The increasing model size is one of the root causes, but another less-emphasized fact is that data scale is actually increasing at a similar speed as model scale, and the training cost is proportional to both of them. Compared to the rapidly evolving model architecture, how to efficiently use the training data (especially for the expensive foundation model pretraining) is both less explored and difficult to realize due to the lack of a convenient framework that focus on data efficiency capabilities. To this end, we present XYZ Data Efficiency, a framework that makes better use of data, increases training efficiency, and improves model quality. Specifically, we propose and combine two data efficiency techniques: efficient data sampling via a general curriculum learning library, and efficient data routing via a novel random layerwise token dropping technique. For GPT-3 1.3B language model pretraining, our work achieves 12.5x less data/time/cost ($3.7K if rent on Azure), while still maintaining 95% of model quality compared to baseline with full data and cost ($46.3K). For GPT-3 1.3B and BERT-large pretraining, our work can also achieve the same model quality with up to 2x less data/time/cost, or achieve better model quality under same data/time/cost. XYZ Data Efficiency is easy to use and tune, enabling us to easily apply it and verify its benefit on additional tasks including GPT-3 MoE model pretraining and small-scale GPT-2/ViT finetuning.

## 1   Introduction

Recently, large-scale deep learning models are empowering us to achieve more in many ways, such as code generation [17] and text-to-image generation [40, 41]. To keep improving the service quality, deep learning model architecture evolves rapidly, and the model size is also growing at a tremendous speed. The increasing model size leads to unprecedented training cost (especially for foundation model pretraining), which recently grows to 2 months on thousands of GPUs/T-PUs [47, 9]. On the other hand, a less-emphasized perspective is that **data scale is actually increasing at a similar speed as model scale, and the training cost is proportional to both of them**. As plotted in Fig. 1, for several

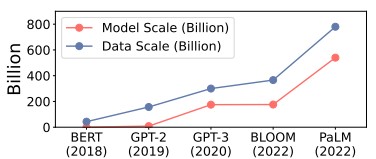

Figure 1: Model scale (number of parameters) and data scale (number of consumed training tokens ) of representative language models in the last 5 years [14, 46, 7, 45, 9].

representative language models in the last 5 years both the model and data scales increase at a similar speed. Recent works including Chinchilla [20] and PaLM 2 [18] emphasize the need of increasing data scale at an even faster speed. This demonstrates the importance of improving data efficiency: achieve same model quality with less data and reduced training cost, or achieve better model quality with the same amount of data and similar training cost.

Submitted to 37th Conference on Neural Information Processing Systems (NeurIPS 2023). Do not distribute.

There are two popular research directions among existing data efficiency techniques: Data sampling techniques aim to improve the convergence speed by sampling the most suitable next data batch from the whole data pool; Data routing techniques aim to reduce the computation by routing each data to only a subset of the model components. These techniques improve data and training efficiency, but existing solutions have several limitations:

- Techniques like curriculum learning improves data efficiency by indexing and sampling training data based on certain difficulty metric [3], and it is recently proved effective on large-scale pretraining tasks [29]. However, implementing different CL strategies for different user tasks can require a lot of code-refactoring, which is time-consuming and error-prone. In addition, existing implementations have less consideration on scalability, which makes it difficult to analyze and index large-scale training data based on different difficulty metrics.
- Existing data routing techniques such as token drop/bypass/pruning were mostly designed for inference and inapplicable to training. TokenBypass [21], to our knowledge the only data routing technique for foundation model pretraining, skips the compute of part of the input tokens at some middle layers during BERT pretraining, reducing pretraining cost while maintaining model quality. However, it requires several special implementations that may only work for the tested BERT pretraining case, such as the importance score-based token dropping decisions and the whitelist for special tokens. This could limit the possibility and benefit of applying it to other cases.
- Although promising data efficiency solutions have been proposed independently, combining multiple methods together for the best outcome is still a laborious process, requiring changes in multiple places in the training pipeline: data loader, data sampler, model architecture, etc. Another challenge is that existing techniques usually add additional hyperparameters but without a clear and low-cost tuning strategy.

To address these above challenges, we present XYZ Data Efficiency, a framework that makes better use of data, increases training efficiency, and improves model quality. Specifically, XYZ Data Efficiency demonstrates the following contributions:

- **Efficient data sampling via general curriculum learning library.** We present a general curriculum learning (CL) library that is both scalable and customizable: it includes a map-reduce based data analyzer that enables scalable analysis and indexing of massive data based on any possible CL metric; it includes a general CL-based data sampler and loader design for users to apply any customized CL strategies. Using this library, we are able to thoroughly explore different CL strategies for GPT-3 1.3B and BERT-large pretraining, and identify the best solution that provides better data and training efficiency than existing CL solution. This library (and the whole XYZ Data Efficiency framework) has been open sourced in a deep learning acceleration library (name hidden for anonymity) that is fully compatible with PyTorch. This will benefit the whole community as a useful tool to apply curriculum learning to their own training tasks.
- **Efficient data routing via random layerwise token dropping.** We present a novel data routing technique called random layerwise token dropping (random-LTD) to skip the computation of a subset of the input tokens at all middle layers. Random-LTD employs a simple yet effective routing strategy and requires minimal model architecture change. It is very flexible to apply random-LTD to various tasks (GPT-3/GPT-3 MoE/BERT pretraining and GPT/ViT finetuning) which the SOTA technique (TokenBypass) does not explore or provides less improvement.
- **An easy to use/tune framework that maximizes data/training efficiency.** XYZ Data Efficiency seamlessly composes the two proposed techniques, and only requires minimal changes on user side. To our knowledge, we are the first to demonstrate that composing data sampling and routing techniques can lead to even better data/training efficiency, especially for foundation model pretraining: For GPT-3 1.3B pretraining, Fig. 2 shows that our approach provides better model quality at all cost budgets, advancing the whole cost-quality Pareto frontier. In particular, we achieve up to 12.5x data/time/cost saving while still maintaining 95% of the model quality (zero-shot eval accuracy) compared to the baseline with full data, while baseline can only maintain 91% of the model quality, a 1.8x higher quality degradation. Based on measured training time, 12.5x would be a cost reduction from $46.3K to $3.7K if renting similar hardware on Azure [2], greatly democratizing research and usage of foundation models for AI community. For GPT-3 1.3B and BERT-large pretraining, we can also achieve up to 2x data and 2x time saving together with better or similar model quality as compared to the baseline training with full data, greatly surpassing state-of-the-art data efficiency solutions as summarized in Tab. 1. Both techniques under our framework are easy to use and tune, and we include a low-cost tuning strategy and a summarized usage guidelines. This enables us to easily apply proposed work and verify its

Table 1: Comparing XYZ Data Efficiency with SOTAs.

| | Efficient data sampling | Efficient data routing | Verified workloads | Key achievements |
|---|---|---|---|---|
| Sequence length warmup [29] | 1 specific CL metric | N/A | GPT-2/GPT-3 pretraining | 1.3x data/cost saving with 100% model quality |
| TokenBypass [21] | N/A | TokenBypass | BERT pretraining | 1.33x data/cost saving with 100% model quality |
| Proposed XYZ Data Efficiency | general CL library support | random-LTD | GPT-3/BERT/MoE pretraining GPT-2/ViT finetuning | 12.5x data/cost saving with 95% model quality 2x data/cost saving with 100% model quality |

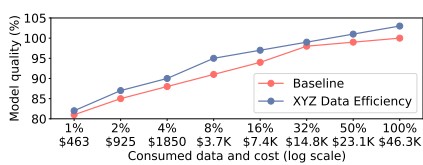

Figure 2: GPT-3 1.3B pretraining: relative model quality (baseline with full data as 100% quality) under different data consumption (1% to 100%) and training cost (when renting on Azure).

benefits on additional workloads including GPT-3 Mixture-of-Experts (MoE) model pretraining and small-scale GPT-2/ViT model finetuning.

## 2 Background and Related Works

**Data sampling.** For deep learning, the most common data sampling method for minibatch stochastic gradient descent is uniform sampling, where at each step a batch of data is drawn uniformly at random from the whole training data. However, it's potentially beneficial to focus on different kinds of data at different training stages. One example is the curriculum learning technique [3] which aims to improve training convergence speed by presenting relatively easier or simpler examples earlier during training. Building a curriculum learning solution usually requires two components: the difficulty metric (i.e., how to quantify the difficulty of each data sample) and the pacing function (i.e., how to decide the difficulty range when sampling next training data batch). In the NLP area, curriculum learning has been applied on small-scale one-stage tasks and downstream finetuning tasks, such as neural machine translation (NMT) [25, 6, 62, 36, 63] and natural language understanding (NLU) [42, 43, 48, 55]. There are also a few works that explore curriculum learning for language model pretraining [37, 61, 8, 29]. However, one common limitation among existing works is that there does not exist a scalable and customizable curriculum learning library, making it difficult to analyze large-scale data and explore custom difficulty metrics/pacing functions. One evidence is that most of the curriculum learning works for language model pretraining only focus on the sequence length metric due to the difficulty of exploring other metrics on the huge pretraining dataset.

**Data routing.** In common deep learning training, the model is considered as a whole and all sampled data will be routed to all model components. However, it's potentially beneficial to route each data sample to only a subset of model components, improving the training efficiency. One direction of efficient data routing is to add data bypassing/skipping capability to existing model architectures such as Transformer. Transformer [49] architecture is a stack of transformer layers, each of which has two main ingredients, i.e., the multi-head attention (MHA) and the feed-forward connection network (FFC). Suppose the transformer has $l$ layers denoted as $L_1, \ldots, L_l$. Let $X_i \in \mathbb{R}^{s \times d}$ be the output tensor of $i-$th transformer layer, and $x_0$ be the input (after embedding) of the transformer. Here $s$ is the sequence length and $d$ is the hidden dimension.

Several token dropping/bypassing/pruning techniques [24, 19, 23, 38, 53] were proposed for BERT inference to reduce the computational overhead, but they are not practical for training. In these works, if a token $i$ ($X_{j,i}$) is decided to be dropped at layer $j$ ($L_j$), the compute cost of this token through all remaining layers ($L_k$ where $k > j$) is eliminated. As such, the sequence length $s_i$ of the $i$-th layer's input $X_{i-1}$ will be a non-increasing array, i.e., $s_0 \geq s_1 \ldots \geq s_l$. However, such a configuration has been shown instability for adaptive token-dropping inference [23]. Therefore, [23] utilize the sandwich rule and distillation from [58] to stabilize training and boost accuracy. But these two methods also significantly increase the training cost. Thus, such techniques cannot be applied to speed up the pretraining procedure.

Recently, TokenBypass [21] enabled token dropping for BERT pretraining. It uses several importance scores/metrics to determine the dropped tokens (token frequency and cumulative loss). It proposed two main mechanisms to overcome the training instability issue: (1) the sandwich token dropping rule, where the first ($L_1$ to $L_i$) and the last few BERT layers ($L_{l-j}$ to $L_l$) capture all tokens (no token dropping) and only bypass $s' \leq s$ tokens from $L_i$ to $L_{l-j}$ middle layers. Particularly, the authors (only) test on the encoder transformer (12-layer BERT$_{base}$ and 24-layer BERT$_{large}$), and let $i = l/2-1$, $j = 1$, $s' = s/2$. (2) special token treatment, where special tokens (e.g., [MASK], [CLS], [SEP]) are never dropped. Compared to TokenBypass, our random-LTD (1) does not require importance score metric, special token treatment, or the sandwich token dropping rule, which dramatically

reduces the manual design effort; (2) has been broadly tested on GPT-3/BERT pretraining tasks and GPT-2/ViT finetuning tasks, providing better data/training efficiency than TokenBypass.

# 3 Design

At high-level, the proposed XYZ Data Efficiency framework has two components as shown in Fig. 3: First we have efficient data sampling, where instead of the baseline's random sampling, we aim to sample the most suitable next data batch from the whole data pool by a general curriculum learning (CL) library. Second we have efficient data routing, where instead of passing all input data to all model components, we aim to efficiently route each data through different components of model by leveraging the proposed random layerwise token dropping (random-LTD) technique. This section presents the design of the two techniques, how we compose them, together with a low-cost tuning strategy and a summarized usage guidelines.

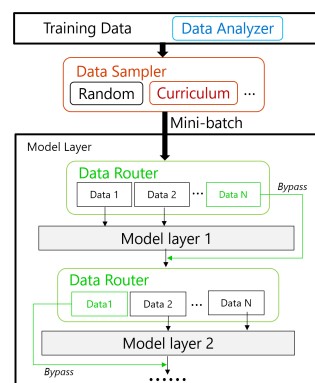

Figure 3: Design of the XYZ Data Efficiency framework.

## 3.1 Efficient data sampling via curriculum learning

To solve the limitations of existing CL solutions as described in previous sections, we design and implement a general curriculum learning library emphasizing the scalability and customizability. It consists of three components as shown in top part of Fig. 3. First we use a data analyzer to perform the offline CPU-only data analysis which indexes the whole data pool based on any difficulty metric, which could be the sequence length, the vocabulary rarity, or anything defined by user. This data analyzer employs a Map-Reduce scheme: During the Map stage, user provides a function that computes the desired difficulty metric, the raw training dataset, and other configurations such as number of CPU nodes and number of threads per node. Then the data analyzer will automatically splits the dataset based on number of workers, compute the difficulty values in a batched fashion, and write the results to two indexes: one index maps each data sample to its difficulty value, and another index maps each distinct difficulty value to the corresponding samples. During the Reduce stage, the data analyzer will merge the index files produced by all workers. This Map-Reduce scheme is necessary since the training data could be huge thus has to be distributed. For instance, we have 173 million data samples (each with sequence length 2048) for GPT-3 pretraining and 2.5 billion data samples (each with sequence length $\leqslant 512$) for BERT pretraining. To reduce the memory overhead when analyzing the huge dataset, we write the index files as numpy memory-mapped files. Using this data analyzer we are able to efficiently analyze GPT-3 and BERT pretraining data based on various difficulty metrics. Using 40 CPU threads on a single node with AMD EPYC 7V12 64-Core Processor, we can finish the analysis on one metric within 3/80 hours for GPT-3/BERT data, respectively.

Next, during training, the curriculum scheduler will determine the difficulty threshold for the current step based on a pacing function such as linear, rooted, or any strategy provided by user. Then the data sampler will sample the data with desired difficulty from the indexed data pool. To apply the proposed CL solution to a existing training pipeline, user just need to call an API and provide the raw training data, the difficulty metric index (computed in the offline analysis), and the pacing function configurations. Our framework will then provide a curriculum learing-based data loader that users can simply iterate at each step. Using our CL library for GPT-3/BERT pretraining, we are able to easily analyze and index the huge training data based on 7 difficulty metrics:

- **Truncation-based sequence length (seqtru), for GPT and BERT.** This metric starts with shorter data samples and gradually increases the sequence length during training. To change the sequence length, this metric will truncate the sequences (from the end of sequence) while keeping the number of samples unchanged, thus the number of tokens will decrease. This metric is recently applied to GPT-2 and GPT-3 models and demonstrate decent training efficiency gains [29].
- **Reshape-based sequence length (seqres), for GPT.** This metric is similar to seqtru metric, but instead of truncating we break the original sequences into segments based on the desired new sequence length. Thus we are essentially "reshaping" the input tensor into more samples and shorter lengths. This metric is proposed in MosaicML Composer as a variant of the seqtru metric [33], but their documentation does not describe which way provides better model quality. We don't apply the seqres to BERT case because unlike GPT data where all tokens are valid, BERT input sequences only include two natural sentences thus each sequence has different "effective sequence length" and then padded to 512. If we simply "reshape" BERT sequences, some of the new short sequences may only contain padding tokens.

- **Reorder-based sequence length (seqreo), for BERT.** This metric is similar to seqtru metric, but instead of truncating we adjust the sequence length by reordering the training data based on the "effective sequence length" in BERT training data sequences.
- **Vocabulary rarity (voc), for GPT and BERT.** This metric was proposed in a CL work for neural machine translation [36]. It computes the product of the unigram probabilities for each sequence by $-\sum_{k=1}^{N} log(p(w_k))$ where $p(w_k)$ is the vocabulary frequency (inside whole training data) of the $k$th word in the sequence. Lower value indicates that the sequence has more common vocabularies.
- **seqtru_voc, for GPT and BERT. seqres_voc, for GPT. seqreo_voc, for BERT.** These 3 metrics are combinations of above metrics. For seqtru_voc and seqres_voc, we first reorder the training data based on voc metric, then apply seqtru or seqres as a kind of post-processing. For seqreo_voc, we treat it as a single new metric and index the data based on it.

Besides the difficulty metrics, another set of CL hyperparameters is the pacing function: the start and end difficulty ($d_s$ and $d_e$), total number of CL steps ($T_c$), and the kind of pacing function (linear, sqrt, or users can plug in any customized function to the proposed framework). For seqtru and seqres metrics, we set the $d_s$ and $d_e$ as value-based (e.g., $d_s = 80$, $d_e = 2048$) since the possible values of these two metrics are continuous. For other metrics, we set $d_s$ and $d_e$ as percentile-based (e.g., $d_s = 1\%$, $d_e = 100\%$) since the possible values of these metrics are discrete. For seqtru and seqres we use a linear pacing function ($d_t = d_s + (d_e - d_s) \times min(\frac{t}{T_c}, 1)$) following the preivous work [29], while for seqreo and voc we use a sqrt pacing function ($d_t = d_s + (d_e - d_s) \times min((\frac{t}{T_c})^{0.5}, 1)$). This is because seqreo and voc will only sample from a subset of data pool before reaching the end difficulty, and previous work finds that in such case it's beneficial to use a sqrt function to avoid sampling too much easy samples at the beginning [36]. Sec. 3.3 includes low-cost tuning strategy and usage guidelines for our CL solutions.

### 3.2 Efficient data routing via random-LTD

**Layerwise Token Dropping.** Existing token dropping methods for inference and training either permanently drop tokens from the compute graph at intermediate layers, or at least make some tokens fully skip a consecutive series of middle layers (Sec. 2). However, several works [50, 31, 51] have shown that MHA focuses on different tokens at different layer depths and the attention map aligns with the dependency relation most strongly in the middle of transformer architectures. Therefore, fully skipping middle layers like TokenBypass [21] may hinder the learnability/generalization of the architecture during pretraining/inference. We conjecture that this might be why multiple first/last layers need to disable token bypassing and the special token treatment is needed.

In order to overcome this problem, we propose a layerwise token dropping (LTD) mechanism. Instead of fully bypassing same tokens over all middle layers, each transformer layer independently drops/retains its own set of tokens. In more detail, recall that the input of $(i + 1)$-th layer ($L_{i+1}$) is $X_i \in \mathbb{R}^{s \times d}$. Denote the dropped token index as $J_i = \{j_1, j_2, ..., j_{a_i}\}$ and the kept token index as $K_i = \{k_1, ..., k_{b_i}\}$ such that $a_i + b_i = s$. We have $J_i \cup K_i = \{1, 2, 3..., s\}$ and $J_i \cap K_i = \emptyset$ for each layer. Meanwhile, for any two different layers $L_{i_1}$ and $L_{i_2}$, $J_{i_1}$ and $J_{i_2}$ are independent, though the dropped ratios are the same. With this layerwise mechanism, each token rarely bypasses all middle layers. Thus, its dependency on other tokens can be captured by MHA.

**Random Token Dropping.** Various importance score-based metrics are used to determine the token dropping criterion. Most of them can be categorized in attention score-based or loss/frequency-based metrics. However, both of them introduce challenges that make LTD less practical: For attention score-based metrics, the compute cost for LTD is too high since the metric has to be calculated for every layer; For loss/frequency-based metrics, the accumulated loss or frequency would not be changed within the same iteration, which leads the dropped token to be the same for different layers, breaking the desired LTD mechanism. Instead of importance score, we propose to use *purely random* token dropping assignment and prove its effectiveness in all our experiments. For each transformer layer, we randomly (uniformly) select a small batch of tokens to proceed with the compute and drop the rest. In more details, assume $M_i = \{m_i(1), m_i(2), ..., m_i(s)\}$ is a random shuffle of $S = \{1, 2, ..., s\}$. Then the dropped token set is $J_i = \{m_i(1), m_i(2), ..., m_i(a_i)\}$ for the input of $L_{i+1}$.

**Random and Layerwise Token Dropping.** Combining layerwise token dropping with random token dropping, we have our final random and layerwise token dropping method (random-LTD), which can efficiently apply token dropping for each individual layer and can capture the attention dependency of each token with other others in middle layers with high probability. As a result, our experiments on BERT pretraining confirm that random-LTD does not require and won't benefit from special token treatment used by the TokenBypass work, further reducing the implementation complexity. Fig. 5

```
1  if meth == "baseline":
2    hs = Layer(hs)
3  if meth == "random-LTD":
4    k_hs, d_hs = gather(hs)
5    k_hs = Layer(k_hs)
6    hs = combine(k_hs, d_hs)
```

Figure 4: random-LTD only requires a few lines of code. hs, k$_{hs}$, and d$_{hs}$ means the full input, kept input, and dropped input. "gather", "Layer", "combine" means the functions for random selection, transformer layer, and order-preserved token combination.

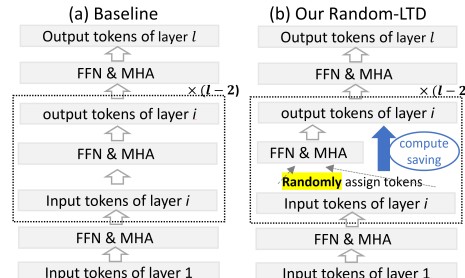

Figure 5: Transformer layers for baseline and random-LTD. The dash-line box is repeated by $l - 2$ times.

presents the comparison between standard baseline training and random-LTD. The pseudo-code is given in Fig. 4. For each layer, random-LTD randomly selects (function "gather") a subset of the tokens and feeds (function "Layer") them into the transformer layer. Afterward, we combine (function "combine") the output of transformer layer with the dropped tokens to recover the full sequence length in a order-preserved manner. Thus, the next layer still receives the full sequence and can repeat this process. To apply random-LTD to an existing training pipeline, user just needs to provide the module class name that they want to apply random-LTD (e.g., a TransformerLayer class). Then XYZ Data Efficiency will wrap the module with a new module that includes token dropping capability, and drop some of the input tokens for this module during training.

**Layers without Token Dropping.** While TokenBypass needs to keep half of the layers in full sequence length training, random-LTD has no such limitation. Thanks to its attention-capture feature, we can apply random-LTD to most of the transformer layers except the first and last layers, enabling further training efficiency gain. Our experiments show that keeping the first and last layers in full sequence length training usually leads to better performance since (1) the first layer directly connects to the embedding, and it can help refine the raw feature; (2) directly connected to the final prediction, the last layer provides a feature realignment for all tokens which could improve the model quality.

**Monotonic Sequence Length Growth.** In order to reduce the gradient variance introduced by random-LTD, we gradually increase the kept sequence length throughout training with a linear schedule (referred to as MSLG). Thus random-LTD has two hyperparameters similar to CL: starting from a sequence length $r_s$ which denotes the size of kept token set $K_i$ for each middle layer after dropping, random-LTD will gradually drop less tokens (following a linear function) and eventually stop dropping after $T_r$ steps. Our experiments show that MSLG provides better model quality than constant drop schedule under similar data/compute savings. Sec. 3.3 includes low-cost tuning strategy and usage guidelines for random-LTD.

### 3.3 Composing CL and random-LTD, tuning strategy, usage guidelines

CL and random-LTD are complementary: CL helps to sample the next data batch, and random-LTD helps to decide how to route each sampled data inside the model. XYZ Data Efficiency hides several complexities when composing the two techniques so that users can easily enjoy the compound benefit. As one example, some CL metrics would affect the actual sample sequence length, thus

Table 2: CL and random-LTD usage guidelines.

| Case | Guidelines |
|---|---|
| GPT-3 pretraining | CL: $d_s$ = 80/1% (seqtru/voc), $T_c$ = 40% of baseline's total steps
random-LTD: $r_s$ = 128, $T_r$ = 70% of baseline's total steps |
| BERT pretraining | CL: $d_s$ = 128/5% (seqtru/voc), $T_c$ = 50% of baseline's total steps
random-LTD: $r_s$ = 128, $T_r$ = 100% of baseline's total steps |
| GPT-2 finetuning | CL: $d_s$ = 32 (seqres), $T_c$ = 70% of baseline's total steps
random-LTD: $r_s$ = 128, $T_r$ = 30% of baseline's total steps |
| ViT finetuning | random-LTD: $r_s$ = 32/66, $T_r$ = 80% of baseline's total steps |

inside our framework we make sure the random-LTD's token dropping mechanism is aware of this, and also adjust the calculation of number of actual consumed tokens which are affected by both techniques. This token consumption calculation is also critical to the learning rate schedule: previous CL work [29] finds that if a CL technique reduces the number of tokens on certain steps, it is desirable to use a learning rate decay schedule based on consumed tokens instead of consumed steps. This is because if baseline and CL use the same step-wise LR decay, it leads to much faster token-wise LR decay for CL which hurts model quality. In this work, we apply the token-based LR decay schedule for both CL and random-LTD. To our knowledge this is the first work to apply such LR schedule to token dropping/data routing techniques, and our experiments show that it does help improving random-LTD's performance. Our CL library's general data analyzer/sampler/loader and random-LTD's module wrapping design makes it easy to apply our framework to different model

training tasks. And the overall composibility of XYZ Data Efficiency enables us to leverage both data efficiency techniques and achieve even better data and training efficiency (Sec. 4).

**Tuning Strategy and Usage Guidelines.** Both CL and random-LTD only have two parameters that need user tuning: the starting CL difficulty/random-LTD seqlen ($d_s/r_s$), and the total CL/random-LTD steps ($T_c/T_r$). [1] And for both CL and random-LTD we find that it's possible to apply a low-cost tuning strategy proposed in previous CL work [29], where we perform binary search on a very small portion (e.g., 2%) of training to find the smallest $d_s/r_s$ and largest $T_c/T_r$ that don't trigger substantial validation loss fluctuations ("whether the perplexity value becomes larger than 1.3x of the previous best perplexity"). For GPT-2 finetuning, given the low training cost we also perform full training of 16 different CL/random-LTD settings which confirm that (1) the low-cost tuning strategy is able to find very good hyperparameters; (2) both CL and random-LTD are not sensitive to hyperparameter choices. Tab. 2 summarizes the usage guidelines based on our tuning results, which we believe can be directly applied to any similar models (at least as a very good starting point for any further tuning).

# 4 Evaluation

We evaluate XYZ Data Efficiency by GPT-3/GPT-3 MoE/BERT pretraining and GPT-2/ViT finetuning. Appendix A.5 includes studies of the TokenBypass method on GPT finetuning and pretraining, further demonstrating the advantages of the proposed random-LTD method.

## 4.1 GPT-3 and GPT-3 MoE pretraining

We use *the Pile* public dataset [16] to perform the pretraining of GPT-3 1.3B [7] (24 layers, 2048 hidden size, 16 attention heads) model. We also pretrain a GPT-3 Mixture-of-Experts (MoE) 6.7B model (24 layers, 1024 hidden size, 16 attention heads, 64 experts on every other layer) following related work [39]. We then perform 0-shot and 10-shot evaluations on 19 tasks to evaluate the model quality of the pretrained models. Detailed experimental setup is described in Appendix A.1.

Among the 5 CL difficulty metrics we have for GPT-3 model, to find out which metric provides the best model quality we pretrain the model (with 100% data) 5 times (each with 1 CL metric). For seqtru metric (to our knowledge the only metric previously applied to GPT-3 pretraining), we tune the CL hyperparameters $d_s$ and $T_c$ based on the tuning strategy proposed in previous work [29]. Then for other metrics we use the same hyperparameters without retuning for fair comparison. As presented in Tab. 3 case 1 to 6, results show that all 5 CL metrics provide better model quality than baseline (except (4)CL_voc's 0-shot accuracy), and the (5)CL_seqtru_voc provides the best quality. The extensibility of our general CL library enables us to easily apply different CL metrics to this large-scale model pretraining with huge training data, and identify a new CL metric that provides better model quality than existing solution (2)CL_seqtru. Next we pretrain the model with 67% data, comparing the baseline and the best CL metric we find. Results show that the average 0-shot evaluation accuracy drops from 42.5 to 41.9 when baseline use less data (Tab. 3 case 1, 9). On the other hand, our CL solution (case 10) with 67% data is able to achieve better 0-shot and 10-shot accuracy than baseline with 100% data, achieving a 1.5x data and time saving.

When applying the proposed random-LTD technique, results show similar benefit as CL: better model quality when using 100% data (Tab. 3 case 7), and 1.5x data/time saving while maintaining model quality (case 11). To explore whether composing CL and random-LTD could achieve even better data and training efficiency, first we pretrain the model with both techniques under 100% training data. Results (case 5, 7, 8) show that using both techniques together further improves the model quality, demonstrating the benefit of composability by our framework. Next we pretrain the model with 50% data. Results (case 12 to 15) show that the baseline has worse 0-shot and 10-shot evaluation accuracy under 2x less data. Using CL or random-LTD can only recover part of the accuracy loss. On the other hand, the composed data efficiency solution is able to achieve the same or better accuracy results as baseline with 100% data, demonstrating a 2x data and 2x time saving.

To better understand how the proposed approach influences the model convergence, Fig. 6 plots the token-wise validation perplexity during pretraining. At the beginning of the training the proposed approach has slower convergence since we focus on easier/simpler data samples (CL) and drop more tokens (random-LTD) at the beginning. On the other hand, at the later stage of training the proposed approach is able to provide faster convergence speed than baseline. Our approach with 50% data is able to achieve similar final validation perplexity as baseline with 100% data (while baseline with 50% data cannot). Our approach with 100% data is able to achieve even better final validation perplexity which leads to the highest model quality.

---

[1]For CL, the ending difficulty $d_e$ is always the highest possible difficulty

Table 3: GPT-3 1.3B (case 1 to 15) and GPT-3 MoE 6.7B (case 16, 17) pretraining cost and average evaluation accuracy on 19 tasks. GPT-3 MoE only has 0-shot accuracy due to time constraints. Accuracy results for each single task can be found in Appendix A.1

| Case | CL/ random-LTD hyperparameter | Data (billion tokens) | Time (hours on 64 V100) | Avg 0-shot accuracy | Avg 10-shot accuracy |
|---|---|---|---|---|---|
| (1)baseline | N/A | 300 (1x) | 260 (1x) | 42.5 | 44.0 |
| (2)CL_seqtru | $d_s = 80, T_c = 110K$ | 300 (1x) | 257 (1.01x) | 43.4 | 44.8 |
| (3)CL_seqres | $d_s = 80, T_c = 110K$ | 300 (1x) | 248 (1.05x) | 43.0 | 44.5 |
| (4)CL_voc | $d_s = 1\%, T_c = 110K$ | 300 (1x) | 257 (1.01x) | 42.3 | 44.5 |
| (5)CL_seqtru_voc | same as (2) + (4) | 300 (1x) | 259 (1.00x) | 43.6 | 44.9 |
| (6)CL_seqres_voc | same as (3) + (4) | 300 (1x) | 248 (1.05x) | 43.0 | 44.4 |
| (7)random-LTD | $r_s = 128, T_r = 200K$ | 300 (1x) | 263 (0.99x) | 43.7 | 44.9 |
| **(8)CL_seqtru_voc +random-LTD** | same as (5) + (7) | 300 (1x) | 260 (1.00x) | **43.8** | **45.1** |
| (9)baseline | N/A | 200 (1.5x) | 174 (1.49x) | 41.9 | 44.0 |
| (10)CL_seqtru_voc | seqtru: $d_s = 80, T_c = 73K$ voc: $d_s = 1\%, T_c = 73K$ | 200 (1.5x) | 171 (1.52x) | 42.7 | 44.5 |
| (11)random-LTD | $r_s = 128, T_r = 133K$ | 200 (1.5x) | 176 (1.48x) | 43.1 | 44.8 |
| (12)baseline | N/A | 150 (2x) | 130 (2.00x) | 42.0 | 42.7 |
| (13)CL_seqtru_voc | seqtru: $d_s = 80, T_c = 55K$ voc: $d_s = 1\%, T_c = 55K$ | 150 (2x) | 129 (2.02x) | 42.6 | 43.7 |
| (14)random-LTD | $r_s = 128, T_r = 100K$ | 150 (2x) | 131 (1.98x) | 42.7 | 43.5 |
| **(15)CL_seqtru_voc +random-LTD** | same as (13) + (14) | **150 (2x)** | **130 (2.00x)** | 42.8 | 44.0 |
| (16)baseline | N/A | 300 (1x) | 111 (1x) | 42.8 | |
| **(17)CL_seqtru_voc +random-LTD** | same as (5) + (7) but with 2x $T_c$ and $T_r$ due to batch size | 300 (1x) | 111 (1.00x) | **43.5** | |

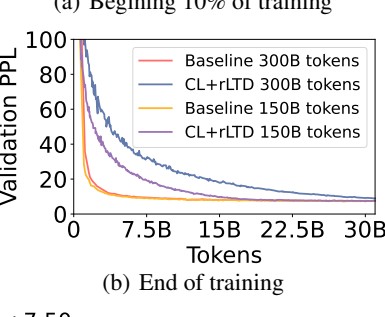

(a) Begining 10% of training

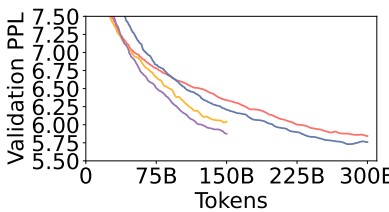

(b) End of training

Figure 6: Validation perplexity during GPT-3 1.3B pretraining, comparing the baseline and the best XYZ Data Efficiency solution under 100% and 50% training data.

As presented in Sec. 1 and Fig. 2, we also compare baseline and proposed work when using even less data during GPT-3 pretraining (Detailed accuracy results can be found in Appendix A.1). Results show that our approach provides better model quality at all cost budgets, advancing the whole cost-quality Pareto frontier. In particular, we achieve up to 12.5x data/time/cost saving while still maintaining 95% of the model quality (zero-shot eval accuracy) compared to the baseline with full data. Based on measured training time, this would be a cost reduction from $46.3K to $3.7K if renting similar hardware on Azure [2], greatly democratizing research and usage of foundation models.

Recent work shows that applying Mixture-of-Experts (MoE) to GPT-style model pretraining could lead to better training efficiency while reaching similar model quality [39]. Thus we also pretrain a GPT-3 MoE 6.7B model (350M base model, together with 64 experts on every other layer) to compare baseline and proposed work. Results show that MoE model does achieve similar model quality with less training cost (Tab. 3 case 1, 16). On the other hand, our approach can further improve MoE model's model quality (case 17), confirming its broad applicability.

## 4.2 BERT-large pretraining

We use *the Pile* public dataset [16] to perform the pretraining of BERT-large [14] (24 layers, 1024 hidden size, 16 attention heads) model. We then perform GLUE finetuning to evaluate the model quality of the pretrained models. Detailed experimental setup is described in Appendix A.2.

Similar to the GPT-3 case, for CL we first investigate which metric (among 5 metrics we have for BERT model) provides the best model quality by pretraining the model (with 100% data) 5 times. Tab. 4 case 1 to 6 results show that 4 CL metrics provide better model quality than baseline, and the (5)CL_seqtru_voc provides the best quality. Next we pretrain with 67% data, comparing the baseline and our best CL metric. Results show that the GLUE score drops from 87.29 to 87.19 when baseline use less data (case 1, 9). On the other hand, our CL solution (case 10) with 67% data is able to achieve on-par GLUE score as baseline with 100% data, achieving a 1.5x data and time saving.

Tab. 4 case 7, 11, 14 present the case when applying random-LTD only. In terms of data saving random-LTD performs better than CL: it is able to achieve better GLUE score even with 2x less data than baseline (case 14), greatly surpassing the 1.33x data saving by the state-of-the-art TokenBypass method. However, the time saving is less than data saving because the token dropping mechanism adds a computation overhead at each step. Because the BERT-large is a smaller model than GPT-3 1.3B, this fixed latency overhead has a larger relative impact to the training time. However, even with this overhead random-LTD is still a more data/time-efficient solution than baseline/TokenBypass.

Tab. 4 case 8 and 15 present the case when applying both CL and random-LTD. At 50% data, the composed solution further improves the GLUE score from the CL/random-LTD-only cases (case 15), achieving a 2x data and 1.8x time saving while maintaining the GLUE score compared to baseline.

Table 4: BERT-large pretraining cost and GLUE finetuning score (median±std, details in Appendix A.2).

| Case | CL/ random-LTD hyperparameter | Data (billon tokens) | Time (hours on 64 V100) | GLUE finetune score |
|---|---|---|---|---|
| (1)baseline | N/A | 1049 (1x) | 261 (1x) | 87.29±0.53 |
| (2)CL_seqtru | $d_s = 128, T_c = 960K$ | 1049 (1x) | 265 (0.98x) | 87.31±0.57 |
| (3)CL_seqreo | $d_s = 5\%, T_c = 960K$ | 1049 (1x) | 261 (1.00x) | 87.48±0.61 |
| (4)CL_voc | $d_s = 5\%, T_c = 960K$ | 1049 (1x) | 261 (1.00x) | 87.36±0.64 |
| (5)CL_seqtru_voc | same as (2) + (4) | 1049 (1x) | 266 (0.98x) | 87.60±0.34 |
| (6)CL_seqreo_voc | same as (3) + (4) | 1049 (1x) | 262 (1.00x) | 87.06±0.52 |
| (7)random-LTD | $r_s = 128, T_r = 2M$ | 1049 (1x) | 302 (0.86x) | **88.17±0.48** |
| (8)CL_seqtru_voc +random-LTD | same as (5) + (7) | 1049 (1x) | 290 (0.90x) | 87.69±0.32 |
| (9)baseline | N/A | 703 (1.5x) | 175 (1.49x) | 87.19±0.49 |
| (10)CL_seqtru_voc | seqtru: $d_s = 128, T_c = 640K$ voc: $d_s = 5\%, T_c = 640K$ | 703 (1.5x) | 178 (1.47x) | 87.29±0.62 |
| (11)random-LTD | $r_s = 128, T_r = 1.34M$ | 703 (1.5x) | 201 (1.3x) | 87.99±0.38 |
| (12)baseline | N/A | 524 (2x) | 131 (1.99x) | 86.61±0.5 |
| (13)CL_seqtru_voc | seqtru: $d_s = 128, T_c = 480K$ voc: $d_s = 5\%, T_c = 480K$ | 524 (2x) | 133 (1.96x) | 86.9±0.33 |
| (14)random-LTD | $r_s = 128, T_r = 1M$ | 524 (2x) | 150 (1.74x) | 87.32±0.48 |
| **(15)CL_seqtru_voc +random-LTD** | same as (13) + (14) | **524 (2x)** | **144 (1.81x)** | 87.44±0.46 |

Table 5: GPT-2 finetuning on PTB results.

| Case | Best PPL at seed 1234 | Num. combinations surpass baseline | PPL median/std over 5 seeds |
|---|---|---|---|
| (1)baseline | 16.077 | N/A | 16.077±0.028 |
| (2)CL_seqtru | 15.888 | 9 out of 16 | |
| **(3)CL_seqres** | 15.795 | 16 out of 16 | **15.818±0.032** |
| (4)CL_voc | 16.031 | 4 out of 16 | |
| (5)CL_seqtru_voc | 16.005 | 3 out of 16 | |
| (6)CL_seqres_voc | 15.981 | 8 out of 16 | |
| (7)random-LTD | 15.910 | 16 out of 16 | 15.948±0.040 |
| (8)CL_seqres +random-LTD | 15.831 | N/A | 15.831±0.014 |

Table 6: ViT finetuning results.

| | CIFAR datasets on 24-layer ViT | | |
|---|---|---|---|
| | Data saving | Top-1 (CIFAR100) | Top-1 (CIFAR10) |
| baseline | N/A | 93.93±0.30 | 99.32±0.05 |
| random-LTD | 1.4x | 94.02±0.40 | 99.30±0.03 |
| | ImageNet datasets on 12-layer ViT | | |
| | Data saving | Top-1 | Top-5 |
| baseline | N/A | 84.65±0.04 | 97.41±0.02 |
| random-LTD | 1.3x | 84.70±0.04 | 97.48±0.02 |

Another thing to note is that this case also has more time saving than the random-LTD-only case. This is because CL will first truncate the sequences before random-LTD perform the random token selection, and the shorter sequences reduces random-LTD's computation overhead. At 100% data, the composed solution (case 8) improves the GLUE score from the CL-only case, but is worse than the random-LTD-only case. One hypothesis is that for BERT pretraining when composing the two techniques it's preferable to reduce the CL duration, but exhaustively testing all hyperparameters is out of our resource budget and this work's scope.

### 4.3 GPT-2 and ViT finetuning

To verify the effectiveness of the proposed work on small-scale tasks, we apply our techniques to PTB finetuning task [30] for an already-pretrained GPT-2$_{350M}$ model checkpoint from Huggingface. Given the much smaller training cost, we focus on improving the model quality under the same amount of data. Detailed experimental setup and hyperparameter tuning are described in Appendix A.3. As shown in Tab. 5, seqres provides the best model quality among the 5 CL metrics (case 3), unlike the two pretraining tasks where the seqtru_voc is the best metric. This is because this finetuning task has much smaller batch size and number of tokens per batch. seqtru will reduce number of tokens per batch, which is less desirable under small-batch training. The small batch also prevents the voc metric to include sufficient number of samples with different vocabulary rarity, limiting its benefit. Applying random-LTD also improves the model quality (case 7). Both CL best metric and random-LTD are able to surpass baseline on all 16 combinations of their hyperparameters, demonstrating that they are not sensitive to the hyperparameter choices. At last we try another 4 seeds for the baseline, CL best metric, random-LTD, and the CL+random-LTD case. The composed CL+random-LTD case (case 8) further improves model quality from random-LTD-only case, but is only on-par with CL-only case. One hypothesis is that for tasks with such small-scale training data, it's less possible to further improve model quality by composing multiple data efficiency techniques.

We also try finetune the vision transformer (ViT) on both ImageNet (with a 12-layer pretrained ViT) and CIFAR (with a 24-layer pretrained ViT). Due to time/resource limitation, we only test random-LTD for this task. Detailed experimental setup is described in Appendix A.4. As presented in Tab. 6, results show that random-LTD is able to achieve 1.3-1.4x data savings while maintaining the model quality, demonstrating its broad applicability.

## 5 Conclusion

Unlike model scale which could reduce in the future with novel architecture, the amount of available training data will increase continuously and irreversibly. Language model pretraining is one of the first to reach a data scale that even training one full epoch is difficult, but sooner or later all machine learning tasks will face the same data efficiency challenge. In this work we propose the XYZ Data Efficiency framework, which demonstrate the power of composing 2 novel data efficiency techniques together. This enables us to achieve an up 12.5x data/time/cost saving (from $46.3K to $3.7K on Azure) while maintaining 95% of model quality for GPT-3 pretraining, an up to 2x saving for GPT-3 and BERT pretraining while maintaining 100% model quality, or to achieve even better model quality under similar data and cost. XYZ Data Efficiency is easy to use and tune, which enables us to apply it and verify the benefit on additional GPT-3 MoE pretraining and GPT-2/ViT finetuning tasks.

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
