# OpenReview forum: "XYZ Data Efficiency: Improving Deep Learning Model Quality and Training Efficiency via Efficient Data Sampling and Routing"
_NeurIPS.cc/2023/Conference — Submitted to NeurIPS 2023_

### Official Review · Reviewer_Na2w · 2023-07-06

**Soundness:** 2 fair
**Presentation:** 2 fair
**Contribution:** 2 fair
**Rating:** 3
**Confidence:** 4

**Summary:**

This work introduces XYZ Data Efficiency, a framework that combines curriculum learning and data routing techniques to improve data efficiency in training recent large models. In detail, authors implemented an efficient difficulty metric calculation method for large datasets by utilizing map-reduce, on top of which authors tried various curriculum learning techniques. Furthermore, by analyzing the limitations of existing data routing techniques, authors developed random-LTD that drops different tokens for different Transformer layers. Finally, authors demonstrate their XYZ Data Efficiency framework leads to achieve the baseline accuracy with less data, or achieve the better accuracy with the same amount of data.

**Strengths:**

1. Developing a general, efficient, and easy-to-use framework for curriculum learning for large models hasn't been explored before to the best of my knowledge. Given the high cost of training recent large models, such library can enable more active research in this field.
2. Their data routing technique (i.e. random-LTD) is thoughtfully designed, and seems to improve the final performance of large Transformers across different tasks.

**Weaknesses:**

This paper touches on multiple aspects of improving training efficiency of large models, especially from the data perspective, but none of them seem to meet the NeurIPS standard.
1. Firstly, I would argue that random-LTD has almost nothing to do with data efficiency. It looks to me that random-LTD is actually closer to some regularization, particularly dropout [1]. Just because one can achieve the same performance with 2x less data with some regularization techniques (e.g. weight decay), calling them as a "data efficiency trick" cannot be justified in my opinion. Since it bypasses some computations of some tokens in some layers, I believe it's closer to a computation efficiency rather than data efficiency trick. From the systems perspective, however, random-LTD consistently hurts the overall throughput as shown in Table 3 & 4 somehow.
2. I doubt the practical utility of map-reduce-based data difficulty calculation. The metrics used in this paper are all offline metrics in that they can be calculated only once before training and can be reused later. While I understand even such preprocessing can take a painfully long time with recent large datasets (e.g. Pile or C4), I don't think the value practitioners will get from this paper would be not so significant. If they can show their framework can be combined with some online or dynamic metrics (e.g. loss value for each token), I would be more convinced.
3. XYZ Data Efficiency framework seems to lack the flexibility and/or modularity, a highly important aspect in the framework. For example, the use of certain CL techniques require specific LR schedules to enjoy the maximal improvement. This essentially means that users get a reduced flexibility in choosing their own LR schedulers. Such entanglement between LR schedulers and data sampling strategies can further harm the user experience when they want to implement their custom data sampling strategies. Overall, my impression is that XYZ Data Efficiency doesn't allow much flexibility for users to try out different things, but rather enforces users to follow their predefined pipeline, in this case, composed of random-LTD and several CL strategies.

To summarize, I find two major framing issues in this paper. First, while CL can be approached from the data efficiency perspective, I believe random-LTD (or data routing) has little to do with data efficiency. Second, I believe XYZ Data Efficiency is more of a combination of two algorithms (i.e. CL and random-LTD) rather than a some general framework due to its lack of flexibility and modularity.

[1] Liu et al., Gating dropout: Communication-efficient regularization for sparsely activated transformers

**Questions:**

1. Can authors provide full train/validation loss curves (from beginning to end)? It's emphasized in the abstract that XYZ Data Efficiency achieves 95% of baseline performance with up to 12.5x less data/time. However, it's generally true that training significantly slows down as it gets closer to convergence. Therefore, it is also possible that the baseline training run also achieves 95% of its maximum performance in the very early stage and takes a very long time to improve final 5%. In this case, comparing time-to-95% would be a more fair metric. Such confusion can be easily resolved by having whole training loss curves.

---

> ### Author Rebuttal · Authors · 2023-08-09
>
> Thank you for your comments and below are our replies.
>
> <Comment 1> "Firstly, I would argue that random-LTD has almost nothing to do with data efficiency. It looks to me that random-LTD is actually closer to some regularization, particularly dropout [1]. Just because one can achieve the same performance with 2x less data with some regularization techniques (e.g. weight decay), calling them as a "data efficiency trick" cannot be justified in my opinion. Since it bypasses some computations of some tokens in some layers, I believe it's closer to a computation efficiency rather than data efficiency trick. From the systems perspective, however, random-LTD consistently hurts the overall throughput as shown in Table 3 \& 4 somehow."
>
> <Reply 1> We agree that the term "data efficiency" could bring confusions. We regarded random-LTD as "data efficient" because some tokens are dropped for a subset of the layers, but it's true that one can also argue that there is still exist layers excluded from token dropping. We believe that it's possible to mitigate this confusion by only calling random-LTD a "data routing method that improves training efficiency". We are also open to any suggestions of how to fix this.
>
> Regarding random-LTD hurting the overall throughput, it's true that random-LTD introduces some computation overhead when performing the token dropping. However, since the token dropping also leads to less computation for a subset of the layers, random-LTD overall is able to lead to substantial computation and time savings while maintaining the model quality as shown in Table 3 \& 4. Thus we believe random-LTD brings far more benefits than its drawbacks.
>
> <Comment 2> "I doubt the practical utility of map-reduce-based data difficulty calculation. The metrics used in this paper are all offline metrics in that they can be calculated only once before training and can be reused later. While I understand even such preprocessing can take a painfully long time with recent large datasets (e.g. Pile or C4), I don't think the value practitioners will get from this paper would be not so significant. If they can show their framework can be combined with some online or dynamic metrics (e.g. loss value for each token), I would be more convinced."
>
> <Reply 2> First of all, we agree that the map-reduce-based data analyzer would provide more benefit for online/dynamic metrics, which we believe is a promising future direction. On the other hand, we believe that the benefit for offline case is still substantial. First, as mentioned in section 3.1, when using 40 threads it took up to 80 hours for our data analyzer to analyze one difficulty metric for the BERT data. Without map-reduce, this would take 133 days to finish in sequential. Furthermore, the amount of data used in training is exploding in recent AI research. In our work, we follow the GPT-3 work in 2020 which used 300B tokens for up to 175B model. Recently, Llama 2 model (arXiv:2307.09288) achieves new SOTA model quality for smaller-scale models, and they used 2 trillion tokens to train up to 70B model. That's 6.7x data scale increase in just 3 years. This trend of exploding data size further proves the necessity of our work.
>
> <Comment 3> "XYZ Data Efficiency framework seems to lack the flexibility and/or modularity, a highly important aspect in the framework. For example, the use of certain CL techniques require specific LR schedules to enjoy the maximal improvement. ...... Overall, my impression is that XYZ Data Efficiency doesn't allow much flexibility for users to try out different things, but rather enforces users to follow their predefined pipeline, in this case, composed of random-LTD and several CL strategies."
>
> <Reply 3> First, we would like to clarify that the proposed methods does NOT require specific LR schedules. Users can use their own LR schedules, which is also needed to better measure the benefit provided by our methods. The only required change, as described in section 3.3, is to change the decay of the used LR schedule from commonly step-based (reduce LR by x after y steps) to token-based (reduce LR by x after y tokens), simply because the proposed methods lead to different amount of consumed tokens in some steps.
>
> Regarding flexibility and/or modularity, first we want to clarify that our work does NOT require users to always compose the two proposed methods. For example, Table 5 shows that for GPT-2 finetuning only using CL actually provides the most benefit. In addition, our methods focus on the data dimension, making them highly compatible with other techniques such as novel model architecture changes and system acceleration techniques. Last but not least, we do not claim that our methods are the ultimate solutions for improving data efficiency. Instead, we aim to create a useful and extensible framework for users to facilitate users to explore and add different data efficiency strategies, which potentially ends up with findings of even better methods.
>
> <Comment 4> "Can authors provide full train/validation loss curves (from beginning to end)? ...... In this case, comparing time-to-95\% would be a more fair metric. Such confusion can be easily resolved by having whole training loss curves."
>
> <Reply 4> In Figure 2 we performed GPT-3 pretraining under a wide range of training budget from 3B to 300B tokens. Results show that the baseline achieves 94\% model quality when trained with 16\% data, while the proposed work achieves 95\% model quality when trained with 8\% data, a 2x data saving. This is consistent with the main results described in the abstract: when no model quality degradation is allowed, our approach can achieve 2x speedup and cost saving; when 5\% model quality degradation is acceptable, the benefit would further increase to up to 12.5x.

---

> > ### Comment · Reviewer_Na2w · 2023-08-19
> > **Thanks for the rebuttal**
> >
> > Thanks for the rebuttal, which resolved my concern regarding Q4. I still have a major concern regarding the "training (or data) efficiency" frame of this paper. More importantly, I don't think this problem can be resolved without a major rewriting. In detail,
> >
> > **Random-LTD**
> >
> > Authors stated that they framed random-LTD as a data efficiency strategy because it drops a fraction of tokes for some layers. In addition, authors provided another interpretation of random-LTD as a "data routing method that improves training efficiency". I unfortunately don't agree with both claims for the same reason I wrote in my original review: random-LTD is closer to a regularization technique that improves "generalization" than "efficiency". As shown in Table 3 & 4, random-LTD consistently leads to improved downstream task performances. In the paper, authors interpret this as "users can achieve the original performance with a reduced cost". However, when users can achieve a better performance with random-LTD than the baseline, why would users suddenly stop training after achieving the baseline performance? To make a stronger argument around efficiency, I believe random-LTD shouldn't lead to a noticeable performance improvement while achieving the baseline performance faster. That being said, even though I believe random-LTD would be a very useful training trick for large models, it still doesn't look like an efficiency trick to me. However, as the storyline of the paper is fully formed around "efficiency", I don't think this issue can be resolved without major rewriting.
> >
> > **LR scheduler**
> >
> > The adoption of CL may require adapting LR scheduler from a step-based approach to token-based approach *to enjoy the improved performance*. This necessarily requires some changes in the code, ranging from modifying the LR update rule inside the schedule class to the call of `scheduler.step()` (maybe something like `scheduler.step(num_tokens)`. When advertised as an "easy to use framework", I don't know how these required code changes are handled within XYZ Data Efficiency. Furthermore, even though it may not be a heavy workload for users, it seems to me the proposed method is closer to a training "trick" rather than "framework".
> >
> > To sum up, even though two proposed methods themselves are interesting, I believe the paper is selling their methods in a misleading way. That being said, I think the paper can be much more improved when their methods are advertised correctly. However, I can't recommend the acceptance for NeurIPS 2023 in its current form, as it will require a large amount of rewriting.

---

### Official Review · Reviewer_8PyZ · 2023-07-06

**Soundness:** 3 good
**Presentation:** 3 good
**Contribution:** 3 good
**Rating:** 4
**Confidence:** 5

**Summary:**

In this paper, the author proposes XYZ data efficiency framework to improve the data/training efficiency in the foundation model training. The proposed framework mainly consists of two techniques, i.e., (1) the efficient data sampling via general curriculum learning library and (2) efficient data routing via random layer-wise token dropping.

**Strengths:**

* The CL-based library is open-sourced and compatible with PyTorch.

* The proposed method achieves considerable training acceleration with minor or no accuracy degradation.

* The author evaluates their method on several different models, including the large language models.


**Weaknesses:**

* Missing the full term of “CL” in the introduction section (line 44). The first explanation shows in line 64.

* The author argues that the previous methods require changing the data loader, data sampler etc. However, the proposed method still needs to change then as well.

* Besides the TokenBypass, there are also several data routing techniques for foundation model training. E.g., [1] [2]. The author should also include those works for discussion and comparison.

[1] EViT: Expediting Vision Transformers via Token Reorganizations. ICLR 2022

[2] Peeling the Onion: Hierarchical Reduction of Data Redundancy for Efficient Vision. AAAI 2023

* I think the previous work [2] also explores the efficiency at the data sample-level and data routing level. So, it is probably not appropriate to claim the proposed method is “the first to demonstrate that composing data sampling and routing techniques can lead to even better data/training efficiency …”

* As the author claims the proposed framework is easy-to-use and admits being open-sourced as one of the contributions of this work, it would be better if the author could submit the anonymous code with the supplementary materials.

* Though the proposed method can achieve considerable overall acceleration, it would be great if the author can provide a discussion about the overhead of data sampling and routing part.


**Questions:**

Please refer to the weakness part.

---

> ### Author Rebuttal · Authors · 2023-08-09
>
> Thank you for your comments and below are our replies.
>
> <Comment 1> "Missing the full term of “CL” in the introduction section (line 44). The first explanation shows in line 64."
>
> <Reply 1> Thank you for catching this and we will make sure to fix this and double check all terminologies in the final version of our paper.
>
> <Comment 2> "The author argues that the previous methods require changing the data loader, data sampler etc. However, the proposed method still needs to change then as well."
>
> <Reply 2> We agree that this part of description did not clearly differentiate the previous methods and the proposed method. Although both of them requires replacing the data loader etc., the advantage of our approach is that it's much easier to analyze the data with different difficulty metrics and sample/load data based on the new metric. For existing methods, applying a new metric would require nontrivial code changes inside different components, while for our approach we generalized the curriculum learning pipeline so that the only requirement is a separate new function to compute the new difficulty metric for each sample. We will improve this part of description in the final version of our paper.
>
> <Comment 3> "Besides the TokenBypass, there are also several data routing techniques for foundation model training. E.g., [1] [2]. The author should also include those works for discussion and comparison. [1] EViT: Expediting Vision Transformers via Token Reorganizations. ICLR 2022. [2] Peeling the Onion: Hierarchical Reduction of Data Redundancy for Efficient Vision. AAAI 2023. I think the previous work [2] also explores the efficiency at the data sample-level and data routing level. So, it is probably not appropriate to claim the proposed method is “the first to demonstrate that composing data sampling and routing techniques can lead to even better data/training efficiency …”"
>
> <Reply 3> Thank you for catching the missing related works and we agree this make it necessary to rephrase the description of our proposed method. On the other hand, we believe that our work provide sufficient contributions beyond these two related works: First, both works only verify their methods on ViT models, while our methods are verified on both NLP and CV large-scale models. Second, both works provide less benefit than our work. When zero model quality degradation is required, the EViT work can only provide 1.15x training speed for ImageNet training (Table 11 in their paper). In contrast, in our work Table 6, our random-LTD method can provide 1.3x training speedup while slightly improving the model quality. The Peeling the Onion work achieves 1.15x training speedup while slightly improving the model quality, which is again less speedup gain than our random-LTD. Furthermore, when combining both of our proposed methods we demonstrate even better training speedup gain.
>
> <Comment 4> "As the author claims the proposed framework is easy-to-use and admits being open-sourced as one of the contributions of this work, it would be better if the author could submit the anonymous code with the supplementary materials."
>
> <Reply 4> The proposed XYZ framework has been open sourced as part of a popular deep learning acceleration framework developed by us (20K+ stars on GitHub). As a result, we find it extremely difficult to anonymize the code. Thus to avoid the risk of violating double-blind policy we could not provide the code during submission. We will definitely include the clear citation to the open sourced code in the final version of our paper.
>
> <Comment 5> "Though the proposed method can achieve considerable overall acceleration, it would be great if the author can provide a discussion about the overhead of data sampling and routing part."
>
> <Reply 5> We agree and will include numerical analysis of the overhead in the final version of our paper. In section 4.2 we did include some discussion of the overhead introduced by random-LTD for BERT-large pretraining. Due to space limit we didn't include further description about other cases where the overhead is more trivial.

---

### Official Review · Reviewer_6DjP · 2023-07-07

**Soundness:** 3 good
**Presentation:** 3 good
**Contribution:** 2 fair
**Rating:** 6
**Confidence:** 3

**Summary:**

This paper draws inspiration from the observation of training costs increasing quadratically with data size, leading to a focus on enhancing data efficiency. To address this issue, the paper presents a framework that optimizes data utilization, improves training efficiency, and enhances model quality. The framework introduces efficient data sampling and data routing methods designed to overcome the challenges associated with data size. Extensive experimental results conducted on various foundation models confirm the effectiveness of the proposed methods, validating their ability to achieve improved data efficiency and overall model performance.

**Strengths:**

1. The paper introduces a framework that combines general continual learning (CL) techniques with random layerwise token dropping for data sampling and data routing. This framework aims to address the challenges of CL by incorporating the random dropping of tokens at each layer, enabling efficient data processing and routing during the learning process.

2. The effectiveness of the proposed method is demonstrated through experiments conducted on various foundation models. The results highlight the remarkable data efficiency achieved by the framework, showcasing its ability to handle continual learning tasks effectively while maintaining high performance with limited data.

**Weaknesses:**

1. While data efficiency is recognized as crucial for various tasks, the paper could provide a more comprehensive study and presentation of how the proposed method enhances models across different data sizes. A more thorough investigation and analysis of the impact of the proposed method on models of varying data sizes would contribute to a deeper understanding of its effectiveness.

2. It is worth noting that the paper's verification of the proposed method is limited to four models. Expanding the experimental evaluation to include a broader range of models would provide a more robust assessment of the method's performance and its applicability across different architectures. This would enhance the credibility and generalizability of the findings.

**Questions:**

1. A direct comparison between the proposed method and Scaling Law [1] is not provided in the paper. It would be valuable to evaluate and discuss how the proposed method differs from or complements the principles and findings of Scaling Law [1] in terms of data efficiency and model improvement.

2. Random layerwise token dropping, as described in the paper, refers to the process of randomly dropping tokens at each layer during the learning process. Drop path [2], on the other hand, is a technique commonly used in neural architecture search where random paths of layers are dropped during training. While both methods involve dropping components during the learning process, how do they differ in terms of the specific units being dropped and affect the performance?

3. The paper primarily focuses on verifying the proposed method's effectiveness on foundation models. It is not explicitly stated whether the method can be applied to tasks with limited data, such as ImageNet. Further investigation and experimentation would be necessary to determine the performance and applicability of the proposed method on tasks with smaller datasets, beyond the foundation models considered in the paper.

[1] Scaling Laws for Neural Language Models
[2] Deep Networks with Stochastic Depth

**Limitations:**

Please refer to the weakness and question part.

---

> ### Author Rebuttal · Authors · 2023-08-09
>
> Thank you for your comments and below are our replies.
>
> <Comment 1> "While data efficiency is recognized as crucial for various tasks, the paper could provide a more comprehensive study and presentation of how the proposed method enhances models across different data sizes. A more thorough investigation and analysis of the impact of the proposed method on models of varying data sizes would contribute to a deeper understanding of its effectiveness."
>
> <Reply 1> We totally agree with the value of testing the proposed methods on more different data sizes. On the other hand, here we would like to summarize the different data scales that we already tested in the paper: First, in Figure 2 we performed GPT-3 pretraining under a wide range of training budget from 3B to 300B tokens, and the proposed method provides consistent model quality gain at most of the budgets, except the smallest 3B which is probably a too small budget for prtraining. Second, in section 4.3 we performed GPT-2 finetuning on PTB and ViT finetuning on ImageNet and CIFAR. These tasks have small data scale, and the proposed methods are still able to provide similar training efficiency and/or model quality gain.
>
> <Comment 2> "It is worth noting that the paper's verification of the proposed method is limited to four models. Expanding the experimental evaluation to include a broader range of models would provide a more robust assessment of the method's performance and its applicability across different architectures. This would enhance the credibility and generalizability of the findings."
>
> <Reply 2> We totally agree that testing the proposed methods on additional models would enhance the credibility and generalizability. On the other hand, our work focused on Transformer models because currently they have the largest size, highest training cost, and highest popularity. Thus we believe that our work has sufficient merit to the general AI research community, substantially lowering the cost and difficulty on researching the Transformer model training.
>
> <Comment 3> "A direct comparison between the proposed method and Scaling Law [1] is not provided in the paper. It would be valuable to evaluate and discuss how the proposed method differs from or complements the principles and findings of Scaling Law [1] in terms of data efficiency and model improvement."
>
> <Reply 3> We believe our work is orthogonal to the findings in the Scaling Law work. In the Scaling Law work, the are 4 key findings related to data: (1) Model performance depends strongly on the size of the dataset. (2) Model size and data size have to be increased simultaneously in order to consistently achieve better model quality. (3) Large models are more sample-efficient than small models, reaching the same level of performance with fewer optimization steps. (4) Under a fixed computation budget, large model with less data can lead to better performance. In our work, Figure 2 reconfirms their finding 1, and our overall experience agree with their other 3 findings. Our proposed methods aim to further improve the data efficiency, but the overall relationship between data and model performance still holds the same. We will add this discussion in the final version of our paper.
>
> <Comment 4> "Random layerwise token dropping, as described in the paper, refers to the process of randomly dropping tokens at each layer during the learning process. Drop path [2], on the other hand, is a technique commonly used in neural architecture search where random paths of layers are dropped during training. While both methods involve dropping components during the learning process, how do they differ in terms of the specific units being dropped and affect the performance?"
>
> <Reply 4> We believe the Drop path work is more like a special case of our work. In Drop path, the whole mini batch are skipped for a subset of layers every time. In our work, we only skip a subset of tokens for each training sample at each layer. We believe Drop path's complete dropping a subset of layers could lead to worse convergence and/or less stable training for the modern Transformer-based models, as we discussed in section 3.2. Furthermore, the TokenBypass work mentioned in our paper also make some tokens fully skip a subset of layers (similar to Drop path), and in appendix A.5 we provided a thorough comparison between random-LTD and the existing work TokenBypass. Results show that random-LTD provides better benefits on both GPT-2 finetuning and GPT-3 pretraining tasks. We will add this discussion in the final version of our paper.
>
> <Comment 5> "The paper primarily focuses on verifying the proposed method's effectiveness on foundation models. It is not explicitly stated whether the method can be applied to tasks with limited data, such as ImageNet. Further investigation and experimentation would be necessary to determine the performance and applicability of the proposed method on tasks with smaller datasets, beyond the foundation models considered in the paper."
>
> <Reply 5> Actually in section 4.3 we did perform ViT finetuning on ImageNet and CIFAR. Results show that the proposed method is able to provide similar training efficiency and/or model quality gain compared to the foundation model pretraining cases.

---

### Official Review · Reviewer_sHP9 · 2023-07-10

**Soundness:** 4 excellent
**Presentation:** 3 good
**Contribution:** 3 good
**Rating:** 7
**Confidence:** 4

**Summary:**

This paper proposes XYZ Data Efficiency, a framework that makes better use of data, increases training efficiency, and improves model quality. The proposed framework features efficient data sampling, efficient data routing, and an easy-to-use practical framework that is integrated into an existing library.

**Strengths:**

- The targetted application on improving the data and training efficiency is an important problem, especially in the era of large models. This paper makes a practical step in this direction.

- I like the writing of the introduction and related works, which provides a structured review and highlights the goal of this paper and its difference from other papers.

- The description of the proposed method is pretty detailed, which can help the reader to have a detailed understanding of how this framework is implemented and part of the time/computation overhead to run this framework.

- Overall, I would say this paper makes good engineering efforts to make the

**Weaknesses:**

- My main concern about this paper is the evaluation observations. Specifically, it seems that under lower-budget training settings (e.g., training with less data and training time), the improvement over the baseline actually shrinks. Will this make this method can only be suitable for teams with a large amount of data and computation resources?

- In the paper, the authors mentioned that previous methods for improving data/training efficiency fail to achieve satisfactory performance under large-scale settings. Is there some numerical evidence for this claim other than the one shown in Table 1?

**Questions:**

Please refer to weakness.

**Limitations:**

The authors may want to address the limitations of this paper in their final version.

---

> ### Author Rebuttal · Authors · 2023-08-09
>
> Thank you for your comments and below are our replies.
>
> <Comment 1> "My main concern about this paper is the evaluation observations. Specifically, it seems that under lower-budget training settings (e.g., training with less data and training time), the improvement over the baseline actually shrinks. Will this make this method can only be suitable for teams with a large amount of data and computation resources?"
>
> <Reply 1> We are not sure which particular results make you feel that the improvement over the baseline shrinks under lower-budget training setting, so here we just try to clarify this with results that demonstrate the opposite: First, in Figure 2 we performed GPT-3 pretraining under a wide range of training budget from 3B to 300B tokens, and the proposed method provides consistent model quality gain at most of the budgets, except the smallest 3B which is probably a too small budget for prtraining. Second, in section 4.3 we performed GPT-2 finetuning on PTB and ViT finetuning on ImageNet and CIFAR. These tasks have small data scale, and the proposed methods are still able to provide similar training efficiency and/or model quality gain.
>
> <Comment 2> "In the paper, the authors mentioned that previous methods for improving data/training efficiency fail to achieve satisfactory performance under large-scale settings. Is there some numerical evidence for this claim other than the one shown in Table 1?"
>
> <Reply 2> Yes we did provide additional numerical evidence other than Table 1. For curriculum learning, as mentioned in section 4.1, the CL\_seqtru case in our results represents the existing work about curriculum learning for pretraining tasks. This existing work is a specialized implementation and cannot be easily modified to explore other curriculum learning. In contrast, our work makes it easy to test different metrics, and the best metric we found did provide better model quality than the existing work in table 3 (case 2 vs case 5), table 4 (case 2 vs case 5), and table 5 (case 2 vs case 3). For random-LTD, (due to the space limit issue) in appendix A.5 we provided a thorough comparison between random-LTD and the existing work TokenBypass. Results show that random-LTD provides better benefits on both GPT-2 finetuning and GPT-3 pretraining tasks.
>
> <Comment 3> "The authors may want to address the limitations of this paper in their final version."
>
> <Reply 3> Yes it's true that we were not able to explicitly address the limitations and impact, partially due to the space limit issue. In terms of limitation, in section 4.2 we mentioned one limitation is that there is one case where composing two techniques provides less benefit than only using rLTD, which might need further investigation in future. Similarly, it'd be helpful to test the proposed methods on other model architectures and training tasks different from what's tested in our paper. In terms of potential negative societal impact, we believe there is no additional impact introduced by our work given the proposed methods focus on improving training efficiency of existing model architecture and training tasks. We will add a "limitation and impact" section in final version of our paper.

---

> > ### Comment · Reviewer_sHP9 · 2023-08-21
> > **Thanks for the rebuttal**
> >
> > I'd like to thank the authors for their rebuttal. I'll keep my rating as it is.

---

### Official Review · Reviewer_NZ9R · 2023-07-26

**Soundness:** 3 good
**Presentation:** 3 good
**Contribution:** 2 fair
**Rating:** 5
**Confidence:** 3

**Summary:**

This paper introduces XYZ, a data sampling and routing framework designed to enhance the efficiency of training large transformer models. XYZ incorporates a user-defined curriculum learning metric for data sampling and leverages token dropping to reduce computational overhead. The authors propose random layer-wise token dropping (random LTD) to efficiently apply token dropping per layer, capturing attention dependencies between tokens in intermediate layers with high probability. The framework's effectiveness is validated through experiments on pretraining GPT-3, GPT-3 MoE, and BERT, as well as finetuning GPT-2 and ViT, achieving up to a 12.5x reduction in data/time/cost.

**Strengths:**

1. This paper introduces the random layer-wise token dropping technique, which demonstrates a novel approach to enhance the efficiency of large transformer model training.
2. The evaluation is across various models of different sizes, including GPT-3, GPT-3 MoE, BERT, GPT-2, and ViT.
3. The paper provides a comprehensive and detailed account of the training setting used in the experiments. Additionally, the authors present a thorough analysis of the results and observations obtained from the experiments.
4. The paper shows substantial efficiency gains using the XYZ framework.

**Weaknesses:**

1. One notable weakness of the proposed framework is its relatively limited performance compared to the baseline when operating at a smaller data scale, as indicated in Figure 6. Further investigation and clarity on the factors contributing to this limitation would be valuable for understanding the framework's practical applicability across various data scales.
2. Despite claims of open-sourcing the XYZ framework, anonymized code or a link to access the implementation is not provided.
3. The paper does not explicitly address the limitations of their proposed framework or discuss any potential negative societal impact.

**Questions:**

1. How does the proposed XYZ framework compare to the baseline when pre-training GPT-3 1.3B on a data set of 75B tokens?
2. Given the observed slower convergence of XYZ compared to the baseline, what are the underlying factors contributing to this behavior? Specifically, does the convergence difference primarily result from the data sampling methodology (curriculum learning - CL) or the token dropping technique (random layer-wise token dropping - rLTD)?

**Limitations:**

The authors have not explicitly discussed the limitations and potential negative societal impact of their work in the paper.

---

> ### Author Rebuttal · Authors · 2023-08-09
>
> Thank you for your comments and below are our replies.
>
> <Comment 1> "One notable weakness of the proposed framework is its relatively limited performance compared to the baseline when operating at a smaller data scale, as indicated in Figure 6. Further investigation and clarity on the factors contributing to this limitation would be valuable for understanding the framework's practical applicability across various data scales."
>
> "Given the observed slower convergence of XYZ compared to the baseline, what are the underlying factors contributing to this behavior? Specifically, does the convergence difference primarily result from the data sampling methodology (curriculum learning - CL) or the token dropping technique (random layer-wise token dropping - rLTD)?"
>
> <Reply 1> First of all, we believe there exist some confusions about Figure 6: The proposed methods' slower convergence in Figure 6(a) does not mean that "the proposed methods have limited performance compared to the baseline when operating at a smaller data scale". This is because no matter how the data scale changes, the best configurations (the number of CL/rLTD steps) of the proposed methods also change in proportion (as summarized in Table 2). Thus no matter how small the data scale/total data budget is, the proposed methods will only have slower convergence at the early stage of that training, yet still provide better final model quality/training efficiency after the full training. The end result will always be similar as Figure 6(b), regardless of data scale. As shown in Figure 2, we did test and demonstrate that the proposed methods provide better final model quality/training efficiency at a wide range of data scales from 3B to 300B.
>
> Second, in terms of which of the techniques contribute more to the convergence slowdown at the early stage of training, our results show that CL contributes more to the slowdown. This makes sense because compared to CL which completely focus on easier data, rLTD still have first and last layer acting as normal layers without token dropping. We will add more figures and analysis about this in the final version of our paper.
>
> <Comment 2> "Despite claims of open-sourcing the XYZ framework, anonymized code or a link to access the implementation is not provided."
>
> <Reply 2> The proposed XYZ framework has been open sourced as part of a popular deep learning acceleration framework developed by us (20K+ stars on GitHub). As a result, we find it extremely difficult to anonymize the code. Thus to avoid the risk of violating double-blind policy we could not provide the code during submission. We will definitely include the clear citation to the open sourced code in the final version of our paper.
>
> <Comment 3> "The paper does not explicitly address the limitations of their proposed framework or discuss any potential negative societal impact."
>
> <Reply 3> Yes it's true that we were not able to explicitly address the limitations and impact, partially due to the space limit issue. In terms of limitation, in section 4.2 we mentioned one limitation is that there is one case where composing two techniques provides less benefit than only using rLTD, which might need further investigation in future. Similarly, it'd be helpful to test the proposed methods on other model architectures and training tasks different from what's tested in our paper. In terms of potential negative societal impact, we believe there is no additional impact introduced by our work given the proposed methods focus on improving training efficiency of existing model architecture and training tasks. We will add a "limitation and impact" section in final version of our paper.
>
> <Comment 4> "How does the proposed XYZ framework compare to the baseline when pre-training GPT-3 1.3B on a data set of 75B tokens?"
>
> <Reply 4> We didn't test the case of 75B tokens, but as shown in Figure 2 we did test the case for 48B (16\%) and 96B (32\%) tokens, where the proposed methods provide better model quality than the baseline.

---

### Decision · Program_Chairs · 2023-09-21

**Decision:**

Reject

**Comment:**

Paper proposes the XYZ framework aimed at enhancing data efficiency for large model training. The framework incorporates known efficient data sampling, leveraging curriculum learning (CL), and proposes a new token dropping methods. Most reviewers highlight the paper's improvement in efficiency in multiple extensive large model experiments and appreciate its claimed open-source availability.

However, concerns are raised about the framing of "efficiency." It is claimed that the framework can achieve 12.5x cost reduction while achieving 95% performance of the baseline. While this is true, this might be misleading to the reader because, from Fig 2, the baseline model itself achieves 95% performance with around 6.25-3.125x cost reduction. There was a discussion of whether token dropping is improving generalization or data efficiency. However, this is not a major issue as generalization improvement is theoretically understood as sample efficiency improvement.

There was limited discussion on limitations of the framework and lack of in-depth study of why and how random layerwise token dropping work. Authors’ candidness in saying that gain comes from CL techniques is appreciated. Experiments show that different CL techniques work best for different settings which makes it harder to use the proposed framework to achieve the cost reductions without extensive costly experiments.

Authors are highly encouraged to address these concerns in their next revision.